# The Effect of Environmental Factors on Mould Counts and AFB1 Toxin Production by *Aspergillus flavus* in Maize

**DOI:** 10.3390/toxins15030227

**Published:** 2023-03-17

**Authors:** Krisztina Molnár, Csaba Rácz, Tamás Dövényi-Nagy, Károly Bakó, Tünde Pusztahelyi, Szilvia Kovács, Cintia Adácsi, István Pócsi, Attila Dobos

**Affiliations:** 1Centre for Precision Farming R&D Services, FAFSEM, University of Debrecen, H4032 Debrecen, Hungary; 2Central Laboratory of Agricultural and Food Products, FAFSEM, University of Debrecen, H4032 Debrecen, Hungary; 3Department of Molecular Biotechnology and Microbiology, Institute of Biotechnology, Faculty of Science and Technology, University of Debrecen, H4032 Debrecen, Hungary

**Keywords:** maize (*Zea mays*), *Aspergillus flavus*, aflatoxin B1, toxigenic isolates, crop year effect, drought stress

## Abstract

The toxins produced by *Aspergillus flavus* can significantly inhibit the use of maize. As a result of climate change, toxin production is a problem not only in tropical and subtropical areas but in an increasing number of European countries, including Hungary. The effect of meteorological factors and irrigation on mould colonization and aflatoxin B1 (AFB1) mycotoxin production by *A. flavus* were investigated in natural conditions, as well as the inoculation with a toxigenic isolate in a complex field experiment for three years. As a result of irrigation, the occurrence of fungi increased, and toxin production decreased. The mould count of fungi and toxin accumulation showed differences during the examined growing seasons. The highest AFB1 content was found in 2021. The main environmental factors in predicting mould count were temperature (T_avg_, T_max_ ≥ 30 °C, T_max_ ≥ 32 °C, T_max_ ≥ 35 °C) and atmospheric drought (RH_min_ ≤ 40%). Toxin production was determined by extremely high daily maximum temperatures (T_max_ ≥ 35 °C). At natural contamination, the effect of T_max_ ≥ 35 °C on AFB1 was maximal (r = 0.560–0.569) in the R4 stage. In the case of artificial inoculation, correlations with environmental factors were stronger (r = 0.665–0.834) during the R2–R6 stages.

## 1. Introduction

Maize is the most widely grown crop in the world [1]. In Hungary, maize has the largest sown area (1 million ha), and it has a significant economic role; maize yield is 6–8 million t year^−1^, of which 48% is exported, 33% is used as animal feed, 18% in the industry, and 1% is used for seed production [2]. The use of maize products is increasingly widespread, with a growing ratio used for animal feed purposes. However, the extreme weather caused by climate change is becoming an even more severe problem for producers each year. According to relevant climate predictions, drier and warmer summers in the Eastern European region will increase the risk of yield reduction, as well as the frequency of the occurrence of toxin-producing fungi [3,4,5,6]. One of these fungi is the *Aspergillus* species, which is the most frequent contaminant of maize. Toxins, especially aflatoxin B1 (AFB1), produced mostly by the *Aspergillus* species, cause the greatest economic damage during the production and storage of maize. They may also have pathogenic and carcinogenic effects on both livestock and humans [7]. 

*Aspergillus* overwinters in soil and on plant residues. The conidia can be transferred by air or vectored by insects to serve as a new inoculum on the host plants [8]. The fungus colonizes and infects maize cobs during flowering through the silk or by means of insect damage [8,9,10], and the colonization continues during the growing season [11]. Dry periods with high temperatures and low relative humidity in the vegetation period support the toxin production by *A. flavus* [12,13]. Mould growth and toxin production are affected by air temperature and humidity, as well as the dynamics of water activity (a_w_) in grains [14]. In the field, under favorable environmental conditions (heat stress, moisture deficit), *A. flavus* becomes competitive against other fungi and continues toxin production after harvest. Giorni et al. [15] examined the growing ability of *A. flavus* and *F. verticillioides* under different environmental conditions (25–35 °C, 0.87–0.98 a_w_). They observed that the nutritional dominance of *A. flavus* was at 30 °C and lower a_w_ in contrast with 20 °C and 0.95 a_w_ of *F. verticillioides*. Sanchis et al. [16] investigated the growth and aflatoxin production by *A. flavus* and A. *parasiticus*. Optimum conditions were 35 °C and 0.95 a_w_ for germination, growth, and toxin production. In a three-year experiment, *A. flavus* and *Gibberella fujikuroi* sp. co-occurrence were observed by Giorni et al. [17]. In years with lower temperatures and more precipitation, AFB1 content in kernels was lower. When the ambient air temperature was higher, and the least rainfall was registered, AFB1 proved to be the highest. The presence of *A. flavus* was negatively related to *G. fujikuroi* incidence.

The toxin-producing *Aspergillus* species cause significant problems in agricultural food production in tropical and subtropical areas, where environmental conditions are optimal for the development of infection and toxin production [18,19]. Several studies have confirmed the presence of aflatoxin-producing fungi and aflatoxin contamination in agricultural areas of Central Europe [3,20,21,22,23]. An increasing number of researchers are dealing with this problem because aflatoxin concentration has recently exceeded the EU limit in several regions [24,25]. The applicable EU limits for aflatoxin B1 in maize are 20 µg kg^−1^ for feed materials, 10 µg kg^−1^ for complementary and complete feeds, and 5 µg kg^−1^ for dairy and young animals (574/2011/EU). For human consumption, the maximum limit is 5 µg kg^−1^ for AFB1 (165/2010/EU). 

The main problem is the presence of toxigenic *Aspergillus flavus* in agricultural areas in Central Europe. Infections have become more common as a result of the extreme weather conditions frequently observed recently [26,27]. Risk maps created by the AFLA-maize model show that the +2 °C and +5 °C scenarios resulted in a significant increase in AFB1 contamination in Central and Southern Europe [28]. For this reason, the *Aspergillus* and aflatoxin problem should be taken seriously also in this region [9,25,29]. In 2007, *A. flavus* occurred at a higher rate, and aflatoxin-contaminated grain was also detected in Hungary. Five years later, in 2012, the aflatoxin contamination rate was also high, and a smaller epidemic was detected in 2017. This pathogen needs warmer conditions for toxin production than propagation. These years were humid and moderately warm; therefore, the fungus was able to infect silks and, subsequently, cobs [30]. Kocsubé et al. [31] isolated and identified a large number of *A. flavus* in the southern part of Hungary, and approximately 25% of the isolates were able to produce aflatoxins. Dobolyi et al. [32] established that the toxigenic *Aspergillus* species are also present in all Hungarian agricultural areas. In 2009 and 2010, *Aspergillus* strains were identified in 63.5% of the investigated fields, of which 18.8% produced more than 5 µg kg^−1^ aflatoxin. As a result of climate change, they represent a potential risk not only in Hungary but also in neighboring countries. In 2012, Kos J et al. [33] observed the same infection in Serbia (the mean level of AFB1 was 36.3 µg kg^−1^). They concluded that hot and dry weather with long dry periods during the growing season favors the development of the infection of maize with *Aspergillus* species. In Croatia, during the 2012 maize growing season, the weather was extremely hot and very dry, which might have favored the AFB1 contamination. The mean aflatoxin B1 value was 81 µg kg^−1^, 38.1% of the samples were contaminated with aflatoxin, and 28.8% contained more than 20 µg kg^−1^ [3]. 

The occurrence of *Aspergillus* species in Hungary, combined with the negative effects of climate change (extreme rainfall and drought) in the Carpathian Basin, may result in an increase of AFB1 in maize. Continuous monitoring of weather conditions and the presence of *Aspergillus flavus* in the field contributes to predicting the aflatoxin contamination in yield and developing prevention strategies and agricultural practices. Therefore, the aim of this research was to investigate the effect of year and environmental factors under natural and artificial contamination, colonization, and toxin production by *A. flavus* in Hungary.

## 2. Results

### 2.1. Kernel Infection 

During the period of analysis, the mould colony counts on non-inoculated (Control) and artificially inoculated (with toxigenic *A. flavus*) ears under rain-fed (NI) and irrigated (I) conditions were detected. Irrigation produced significant differences (*p* < 0.05) in mould CFU g^−1^ in the three examined years (Table 1). Mould concentration positively correlated with irrigation for both control and inoculated plants. The fungal occurrence in irrigated plots was almost double that in non-irrigated ones. Under natural conditions, the infection by *A. flavus* is also influenced by environmental conditions (temperature, precipitation, humidity) and the presence of insects and competitor fungi. In the case of artificial inoculation, the isolates are embedded under the husks, where the conditions (temperature, a_w_) are optimal for their further development. In the control treatment, the number of mould colony-forming units was between 3.3 × 10^4^ CFU g^−1^ and 4.7 × 10^6^ CFU g^−1^, and, in the case of artificial infection, these values were between 4.5 × 10^5^ CFU g^−1^ and 1.7 × 10^7^ CFU g^−1^, reflecting the success of contamination with *A. flavus*.

Significant differences were found between the examined years in the change of fungal incidence (Figure 1). In the non-irrigated control treatment in 2020, the fungal incidence was significantly higher than in 2021 and 2022 at the *p* < 0.1 significance level. Under irrigated conditions, there was no significant difference between the three years.

In the case of contaminated plants, the mould colonization of maize cobs at rain-fed water supply was significantly higher in 2021 and 2022 compared to 2020 (*p* < 0.05). In the irrigated plots, 2021 was an outstanding year compared to the other two years, with an average CFU value of nearly 12 million g^−1^.

### 2.2. Aflatoxin Contamination

Significant differences were found between the various irrigation treatments in control (non-inoculated) and inoculated cobs (Table 2). Mycotoxin production of *A. flavus* increased significantly under dry conditions. The AFB1 content in kernels was considerably higher in non-irrigated plots in both treatments. In artificially inoculated ears, the amount of toxin was measurable using different water supply regimes. In non-irrigated ears, the toxin concentration was 192.31 µg kg^−1^, while in the irrigated ones, it was 77.42 µg kg^−1^.

Regarding the amount of AFB1, significant differences were found between the examined years (*p* < 0.05, *p* < 0.01). Under natural conditions, AFB1 concentration was low, below the EU limit (5 µg kg^−1^) in all three years (Figure 2a). In the case of the artificial inoculation strain, the toxin content of the kernels exceeded the EU limit every year. In both water supply regimes, AFB1 contamination was the highest in 2021, reaching 463.1 µg kg^−1^ in the non-irrigated treatment and 163.8 µg kg^−1^ in the irrigated treatment (Figure 2b).

The correlation between fungal contamination and AFB1 content was examined during the three years (Figure 3). Linear, logarithmic, and polynomial regression was used to find the highest correlation between variables. Based on the results, 17.66–26.34% of the fungi were able to produce aflatoxin.

### 2.3. Seasonal Characteristics of Climatic Parameters

During the vegetation period, the reproductive stages of maize development stretched from late July to early- or mid-September in the examined years. In the recent growing seasons, the course of summer rainfall tends to become immensely uneven. The R2-R6 period in 2022 showed the slightest (5.6 mm, −8.5%) negative anomaly of all three studied years, while the entire growing season of 2022 (from May to the end of the R6 stage) was 132.3 mm drier (−53.3%) than the 30-year normal (1991–2020), resulting in the most severe drought in the area for decades. In contrast with 2022, rainfall anomalies during the R2-R6 period reached −30.2 mm (−42.5%) in 2020 and −50.2 mm (−58.4%) in 2021. Overall, the rainiest growing season was observed in 2020 with +53.4 mm (+19.9%), and, similarly to 2022, it was extremely dry in 2021 with −110.1 mm (−41.0%).

Correspondingly, the most severe atmospheric drought occurred between the R4 and R6 stages (mid-August to mid-September) of 2021, marked by 7, 6, and 9 days in each stage, respectively, with daily minimum relative humidity lower than 40%, registered per stage. In 2020, the humidity was far from extreme, while in 2022, drought stress proved to be frequent in R3, R4, and R6. The highest duration of relative humidity above 70% and 90%, which might have supported fungal growth significantly, occurred in the R4 and R6 stages in 2020, in R5 and R6 in 2021, and in R5 in 2022.

The longest dry state of the leaf wetness parameter was observed during the reproductive phase in the R5 and R6 stages in 2020 and 2021, indicating the longest periods without rainfall and dew formation. Despite seasonal cooling and longer nights, hence the increasing frequency of nighttime condensation from late August to mid-September, the highest durations of dry state were generally measured in R6. It is essential for a better interpretation of the obtained data that R5 and R6 tend to be by far the longest stages of the reproductive phase of maize growth. Depending on the growing degree days (GDD) accumulation rate, the R5, and R6 may be 2–3 times the length of R1, as observed in 2020 and 2021. Expressed in percentage of growth stage length, the dry state gradually decreased from 80–90 to 55–60 from R2 to R6, with occasional drops to as low as 42–45%, as recorded in R4 2020 and R5 in 2022.

Another crucial parameter regarding plant and fungal development, similar to humidity, is heat conditions. The highest average temperatures were registered in the R2-R4 stages in 2021, exceeding 25 °C in R2. The number of heat stress days, with a daily maximum temperature (T_max_) higher than 35 °C, was as high as 3 in R4 during the growing season, in contrast with 2020, when the temperature never reached 35 °C. In 2022, in addition to the 2 days of T_max_ over 35 °C, 26 and 17 days were registered with T_max_ exceeding 32 and 30 °C, respectively. The relatively large number of heat days was evenly distributed throughout the reproductive phase from R2 to as late as R6 (mid-September). Table 3 shows a cumulated series of temperature, humidity, rainfall, and leaf wetness data.

### 2.4. Relationships between Mould Colonisation, Toxin Production of A. flavus, and Environmental Parameters

Based on the obtained regression analysis, changes in mould count (Figure 4) and aflatoxin B1 contamination were illustrated (Figure 5) in response to environmental factors measured at the two height levels (lower: 0.3 m, upper: 1.5 m AGL) in the plant canopy.

During the reproductive phase of maize growth, the strength of the relationship between the climatic parameters and the mould count was found to be varying with time and treatment. In the case of maximum temperature, Pearson’s correlation coefficient (r) ranged from −0.4 to 0.6, with a fairly uneven course. As an exception, the number of T_max_ ≥ 35 °C days showed a steady, medium-strong relationship with mould count, with a large difference between the control (zero to slightly negative r) and the inoculated (pronounced positive r) treatments. Similar (r ≈ 0.4–0.6) values were found with the inoculated treatments in the case of the daily mean temperature; however, the correlation changed to slightly negative from R5. With the number of days with minimum values of relative humidity ≤ 40%, r slightly exceeded 0.5 in the inoculated treatments, although the correlation was initially negative in the upper canopy zone in the R2 and R3 stages. The length of humid periods, represented by the duration of RH ≥ 70% and 90%, typically showed a weak positive or zero relationship in the control groups and a weakening negative correlation in the inoculated treatments over the R2-R6 period (Figure 4).

Regarding the correlation of the environmental parameters with the aflatoxin content, it was generally stronger than that of the mould count. The highest r values were found in the case of the number of T_max_ ≥ 30 °C and T_max_ ≥ 32 °C days (in R2 stage r ≈ 0.4–0.7), while T_max_ ≥ 32 °C days proved to be an outstanding predictor with r values ranging from 0.5 to as high as 0.8 for the entire R2–R6 period. Inoculated treatments showed a slightly higher correlation with temperature-based parameters in comparison with the control, underlining the significance of inoculation. The relatively small differences imply, however, that the natural presence of *A. flavus* may as well result in serious AFB1 contamination in a hot environment. Over time, the weight of daily mean temperature as a predictor decreased and eventually turned to a negative relationship with toxin content in the inoculated treatments, while it remained mostly neutral in the control groups. The lack of moisture on the canopy surface, i.e., the dry state of leaf wetness, had the strongest effect on toxin production towards the end of the reproductive phase despite an initial (R2) neutral period. The relationship between the number of low (≤40%) minimum relative humidity days and AFB1 concentration changed from negative (upper canopy levels) or weakly positive (lower canopy levels) to medium to strongly positive. In accordance with the expectations, the total duration of time that maize spent under high (RH ≥ 70% and 90%) humidity was in a negative correlation with toxin production in the R2–R5 stages. The strength of the correlation gradually decreased, and the parameter became neutral by the end of the R6 stage. In summary, humidity supported plant health rather than toxin production during the first half of the reproductive phase, even though the effect of atmospheric drought was limited to the lower part of the canopy. During R5 and R6 stages, very dry periods seemed to promote toxin production in all treatments, while the longer humid periods did not have a noticeable effect (Figure 5).

Multiple linear regression was used to explain the effect of environmental factors on kernel infection and aflatoxin B1 accumulation. The strongest correlations and the climatic parameters were determined to predict the incidence of fungi (CFU g^−1^) and mycotoxin amount (AFB1). These variables significantly contributed to prediction, *p* < 0.05.

Due to the observed low correlations, the control treatment and inoculated R3 phenophase at a low canopy level were excluded from the regression in the case of the mould count (CFU g^−1^) (Table 4). After the infection, based on the model results, the fungal incidence is affected by the following meteorological factors: T_avg_, T_max_ ≥ 30 °C, T_max_ ≥ 32 °C, T_max_ ≥ 35 °C, RH_min_ ≤ 40%, RH ≥ 70%, and RH ≥ 90%. The correlations between ecological parameters and mould growth were moderately positive. In the phenophase after inoculation, air temperature determined the development of fungi, then after the R5 phase, air dryness was the main influencing factor.

In this part, the effect of environmental parameters on AFB1 toxin production was investigated in the different phenophases after artificial contamination (Table 5). In the case of the control cobs, temperature determined the AFB1 toxin concentration at both measurement levels. Correlations between mycotoxin amount and T_avg_ and T_max_ ≥ 35 °C were moderate (r = 0.475–0.579). In the R4 stage at the lower and upper measurement levels, the effect of T_max_ ≥ 35 °C on the amount of toxins was maximal.

Even in the case of artificial contamination, the air temperature was the main influencing factor of mycotoxin production, but in the lower part of the canopy, microclimatic conditions also had an effect on AFB1 production due to water supply, strengthening the correlations between the observed parameters. At the lower measurement level, the Pearson correlation is highest in the R4 phenophase (r = 0.763–0.834). At the upper measurement level, in the initial stage of kernel development, AFB1 amount was determined by average air temperature (T_avg_), then T_max_ ≥ 35 °C, and, at the end of the growing season, leaf wetness duration. Near the cobs, in the R2 phenophase, the effect of average temperature (T_max_) and relative humidity (RH ≥ 70) is the strongest (r = 0.834); in the R3 stage, the daily maximum temperature ≥35 °C is the main influencing factor (r = 0.788).

## 3. Discussion

In a three-year-long experiment, the effect of environmental parameters on mould contamination and aflatoxin B1 production of naturally occurring and artificially introduced *A. flavus* was examined. In this study, the amount of AFB1 in the control treatment was below the EU limit (5 µg kg^−1^), as seen in Figure 2a. Dobolyi et al. [32] confirmed that toxin-producing *Aspergillus* strains were already present in fields in Hungary in 2009–2010. *Aspergillus* strains were found in 63.5% of the examined samples, of which 18.8% were able to produce aflatoxin. In 2012 and 2017, aflatoxin-contaminated grain was also detected [34]. Virág et al. [35] examined the presence of *A. flavus* and *A. niger* in the field between 2019–2020. They established that the pathogens occurred in both years and were adapted to the Hungarian climate. Several researchers emphasize the importance of detecting the geographical spreading of toxigenic fungi and the development of models to analyze the effect of environmental factors [28,36,37]. 

Based on the average of the three examined years, the numbers of the colony increased as a result of irrigation (Table 1). In non-irrigated treatments, dry periods due to the lack of rainfall and extremely high daily maximum temperatures increased the production of aflatoxin B1 by *A. flavus*. Kebede et al. [38] evaluated the relationships between toxin production by *Aspergillus* species and plant physiological responses under irrigated and non-irrigated conditions. They found that in 2009 the toxin contamination slightly increased in the non-irrigated ears, and in 2010 the amount of aflatoxin in maize kernels was significantly higher in the irrigated than in the non-irrigated treatment. Similarly, previous studies [39,40] showed that the amount of AFB1 in kernels was higher in non-irrigated than in irrigated plots. Irrigation of maize helps to reduce drought stress, potentially leading to reduced aflatoxin levels in kernels [41]. Jones et al. [40] investigated *A. flavus* infection and aflatoxin B1 contamination in irrigated and non-irrigated maize. Based on their results, the level of contamination and toxin production decreased as a result of irrigation in drier growing seasons, which are one of the major risks of climate change in Hungary.

Drought and heat stress during the growing season of maize have a critical role in aflatoxin B1 contamination. The crop years, characterized by the data series of the daily maximum and mean temperature, daily minimum relative humidity, the duration of humid periods, and dry leaf wetness state and the precipitation, significantly influences the occurrence and development of fungi and the toxin production by *A. flavus*. In this experiment, the amount of aflatoxin was the highest in 2021 in artificially contaminated ears (Figure 2). The number of heat stress days, with a daily maximum temperature (T_max_) higher than 35 °C, was the highest in this year, and the daily minimum relative humidity regularly dropped below 40% between the R3 and R5 phenological stages. The high temperature was the most important environmental parameter contributing to the pre-harvest colonization of maize by *A. flavus* and aflatoxin B1 contamination in kernels. Artificial inoculation has a higher rate of *A. flavus* infection than natural infection when rainfall is low, and the temperature is high during the flowering period. Inoculation has an important role in AFB1 production, too. The amount of the toxin is significantly higher in inoculated ears than in the naturally infected control [17]. Kernel infections were significantly higher in the inoculated treatment than in the non-inoculated control [42]. Plant water stress combined with persistent hot periods supports the colonization and development of the fungus [43]. Camiletti et al. [44] reported that the levels of natural infection by *A. flavus* changed with the drought-stress period in the growing season. 

In the case of the control treatment, temperature determined the toxin concentration, and correlations between mycotoxin amount and T_avg_ and T_max_ ≥ 35 °C were moderate. In the case of artificial infection, temperature proved to be the main influencing factor, as further confirmed by other works. In the initial stage of kernel development, the AFB1 amount was determined by the average air temperature, later T_max_ ≥ 35 °C, and at the end of the growing season, leaf wetness duration (Table 5). Based on the results of Burns and Abbas [45], if the ambient temperature was higher than 32 °C for 41 days during maize kernel filling, a higher aflatoxin contamination level was measured than in the following years, in which the ambient temperature was higher than 32 °C for 30 days. Diener et al. [46] found that the optimal conditions for developing fungi are 36–38 °C, with high humidity of over 85%. In correspondence with the obtained results of our research, Windham et al. [47] found that high maximum daily temperatures showed a positive correlation while rainfall, i.e., the lack of drought stress, was negatively correlated with toxin production.

In summary, our results further confirmed that the most widely accepted consequences of climate change, i.e., the increasing frequency of hot and dry periods and the uneven timely distribution of precipitation resulting in serious drought periods during the growing season, play an important role in the increasing AFB1 contamination, in addition to the limiting effect on the yield of rain-fed maize. Extreme temperatures, drought, and unequal rainfall distribution caused by climate change may stress plants and promote *Aspergillus* infection [48]. Battilani et al. [28] predicted aflatoxin contamination with a future temperature increase of +2 °C and +5 °C caused by climate change. It has been established that even a +2 °C scenario will cause food safety problems in maize in Europe for the next years. Extreme weather conditions are responsible for the decrease in yield and the development of toxin infection in maize [3,16,20,38,39].

## 4. Conclusions

According to the findings of this study, if *A. flavus* infects maize cobs during silking, its toxin production is triggered by high temperatures and atmospheric drought between the R2-R6 stages. As a result of an enhanced crop water supply, fungal growth was promoted, while the toxin content in the kernels decreased, highlighting that irrigation is a solution for farmers to prevent the toxin production of the fungus. Increasing temperature and extreme weather anomalies caused by climate change favor the colonization, development, and toxin production of the *Aspergillus* species. Predictions suggest that this phenomenon could potentially pose a problem in the Central European region in the future. Further field studies are essential to be carried out to reveal the presence of toxigenic *Aspergillus* strains and to develop crop management solutions in order to reduce toxin production. 

In the continuation of our research, it is planned to increase the quantity of samples (repetitions) and/or repeat analyses when a longer data series is available. The set of predictors is as well planned to be further selected and refined, with new environmental parameters as independent variables to be tested and included in order to minimize the effects of collinearity. 

## 5. Materials and Methods

### 5.1. Field Experiment

The effect of irrigation and meteorological parameters on the development of *Aspergillus flavus* and aflatoxin contamination in maize was evaluated. The experiment was carried out during the 2020/2022 growing season on chernozem soil in Hungary at the Experimental Station of the Centre of Agricultural Sciences, University of Debrecen. A complex small-plot field experiment was conducted where the SY Orpheus (FAO 370–390) hybrid was evaluated. The main plot treatments were two irrigation levels, and the sub-plot treatments were three levels of N-fertilizer. The three levels of N were 60, 120, and 180 kg N ha^−1^. In this paper, the fertilizer treatments are not taken into account because, in our experiment, they had no significant effect on toxin production. Each plot (100 m^2^) contained 12 rows, with 76 cm row spacing and 20 cm spacing between plants per row. Seeds were sown manually. 

Two water supplies were used: I (Irrigated) represented the optimum water supply plots where the loss of evapotranspiration (ET 100%) was replenished; in non-irrigated plots (NI), the plants were grown under rain-fed conditions. The loss of evapotranspiration was determined by the daily weather data from a weather station near the experiments, using the method described by Shuttleworth and Wallace [49]. The calculation of the required irrigation water was carried out based on actual evapotranspiration, as described by Allen et al. [50]. The drip irrigation system was applied, and the amount of irrigation water was 60 mm in 2020, 80 mm in 2021, and 235 mm in 2022.

### 5.2. Measurement of Environmental Parameters

Within the experimental plots, air temperature, relative humidity, and leaf wetness duration were logged continuously as basic information on the environmental conditions that affect fungal growth and toxin production. In addition, rainfall and irrigation data were also logged during the three examined growing seasons studied.

Basic climatic data were collected with a standard weather station deployed outside the plots at a distance of ca. 50 m from the experiment. In addition to the climatic data representative of mesoclimatic conditions, the microclimate of the planted plots was aimed to study. For this purpose, a wireless mesh of sensors and a data logger were set up within the field plots to log the below detailed environmental parameters during the relevant maize growth stages.

#### 5.2.1. Temperature and Humidity

Air temperature and relative humidity data were collected in each experiment plot. To reach a relatively high spatial resolution, Onset’s (Onset Computer Corporation, Bourne, MA, USA) RXW-THC-B-868 wireless combo sensors were applied to spare cable work and avoid losses due to possible cable damages in an actively cultivated plot. Both temperature and relative humidity were measured at two heights at each location. The lower level at 0.3 m represents the section of canopy where the soil is the main driver of microclimate, and the spread of spores of wild strains from the soil takes place by splashing water. The upper level at 1.5 m was chosen as the average height of ears and maximum leaf area density.

#### 5.2.2. Duration of Leaf Wetness

Leaf wetness data is interpreted as a time period during which water is present on the surface of the plant canopy. The duration of a completely dry canopy surface, as an indicator of drought, i.e., the lack of any atmospheric water source (rainfall and dew formation), was also continuously logged and summed daily. The wetness state has been measured with PHYTOS-31 type dielectric sensors (METER Group Inc., Pullman, WA, USA). Mounting heights were the same as those of the temperature and humidity sensors.

#### 5.2.3. Precipitation

Daily rainfall was measured outside the experimental plots to avoid errors due to canopy interception. Precipitation was collected using a Campbell Scientific (Logan, UT, USA) ARG-100 tipping bucket-type rain gauge. (Technical details of the sensors for all examined parameters are presented in Table 6.) The applied irrigation doses were added to the natural precipitation for the respective treatments.

### 5.3. Post-Processing and Evaluation of the Data Series

With daily aggregation from the original 5-min resolution data series, minimum, maximum, sum, and mean values were extracted. Since plant and fungal stresses that affect the fungal life cycle and toxin production are closely related to the extremes in temperature, humidity, and water supply, emphasis was placed on the severity and frequency of their occurrence. All derived parameters were calculated for 190 growing degree days (GDD) long reproductive (R2—Blister, R3—Milk, R4—Dough, R5—Dent, R6—Physiological maturity) stages of maize [51,52,53] and finally transformed into a cumulated data series.

Calculation of GDDs for maize:GDD = ∑(T_avg_ − T_b_)
where T_avg_ is the daily mean temperature, and T_b_ is the base temperature for maize set to 10 °C in accordance with most sources [54,55].

#### 5.3.1. Temperature and Relative Humidity

As basic parameters, mean values were calculated for both air temperature and relative humidity. To obtain a more valuable indicator of the degree of severity of extreme heat, the total count of the days with a maximum temperature equal to or higher than 25, 30, 32, and 35 °C was determined. Similarly, the magnitude of the atmospheric drought was studied through the count of days with minimum relative humidity equal to or below 40, 30, and 20%.

Since simple average values tell little about how suitable atmospheric humidity conditions are for fungal growth, the duration of relative humidity equal to or higher than 70 and 90% has also been computed. The parameter is given in days for the respective R stages.

#### 5.3.2. Leaf Wetness State

The duration of the state of the wet canopy and the completely dry canopy was summed daily and, as a next step, for each R stage. The obtained results can be interpreted as a sum of durations of wet and dry states during the given growth stage (5–17 days long each) in days.

#### 5.3.3. Precipitation

Simple sums of daily rainfall were calculated for the respective growth stages, with the irrigation doses added where applied.

### 5.4. Aspergillus Flavus Artificial Contamination

The fungal isolate that was used in the micro-plot experiment was cultivated on Petri dishes, and spore suspensions (10^6^ mL^−1^) were produced. The treatment of plants was carried out through a puncture canal on 12 August 2020, 2 August 2021, and 4 August 2022 in the blister (R2) phenological stage. Inoculations were carried out with the toxinogenic *A. flavus* DE 416/4 strain. For the control treatment, each ear was marked for subsequent identification.

Reaching the final stage, samples were collected from the plots by extracting the marked inoculated and control ears. The ears were delivered to the laboratory separately for each treatment in paper bags. Here, after the peeling, the obtained maize samples were dried in a drying cabinet (LABORMIM Ltd., Budapest, Hungary) at 60 °C ± 1 to ensure weight stability. As a next step, maize samples were crumbled and ground. 

### 5.5. Total Mould Count Determination

Dried kernels were grounded and collected in sterile Stomacher homogenizer bags, suspended in a 1:9 ratio with Buffered Peptone Water (BPW) solution (Scharlab, Barcelona, Spain), and homogenized with a Stomacher Masticator homogenizer (IUL Instruments, Barcelona, Spain) for 2 min, repeated twice. As a next step, decimal dilutions were made of the suspensions, and total mould counts were determined on CYG Agar (Scharlab, Barcelona, Spain) medium, applying the pour plate method. Inoculated solid agar media were incubated at 30 °C for five days for mould count determination. All inoculations were done in triplicate.

### 5.6. Detection of Aflatoxin B1 with HPLC-FD

All HPLC measurements were performed on Dionex Ultimate 3000 (Thermo Scientific, Waltham, MA, USA) HPLC equipment. Dried samples (25 g) were homogenized with 2.5 g sodium chloride (VWR) and 50 mL of 80% methanol (HPLC, Sigma-Aldrich, St. Louis, MO, USA) at high-speed mixing. The extract was diluted in a 1:4 ratio with 40 mL of distilled water. The homogenized sample was filtered into spherical flasks through filter paper (Macherey-Nagel). The diluted extract was filtered, and 10 mL of it was loaded onto the aflatoxin immunoaffinity column (VICAM AflaTest WB HPLC Columns, Weber Consulting Ltd., Göd, Hungary). The column was washed with 10 mL of distilled water, and the toxin was eluted with methanol (5 mL) and was evaporated in Rotavapor R114 (Büchi). After the addition of 1 mL mobile phase (methanol: water, 45:55), the solute was filtered through Millex-GV 0.22 μm filter (Merck-Millipore) and applied to HPLC. Phenomenex (Torrance, CA, USA) RP-C18 column (150 × 4.6 mm, 5 µm) was used with Romer UV derivatization unit (Romer Labs Ltd., Tulln, Austria) and a fluorescence detector ex360 nm, em440 nm, with methanol: water (45:55) eluent. Biopure Aflatoxin Mix 1 standard solution (Romer Labs, Tulln, Austria) was applied to the column.

### 5.7. Statistical Analysis

Analysis of variance (ANOVA) was performed with SPSS 28 statistical software, and the differences between means were compared with the Duncan’s Multiple Range Test and Independent-Sample T-test at the 0.05 probability level. To explain the changes in the amount of fungi and aflatoxin accumulation, linear and polynomial Pearson’s correlation was used. Multiple regression analysis was carried out to reveal the relationships between the colonization and toxin production of *Aspergillus flavus* and climatic parameters in the different phenophases of maize. Data of mould count (CFU g^−1^) were lg transformed, while aflatoxin content (AFB1) was ln transformed before statistical analyses to reduce the variance of the original data [56]. The transformed data had a normal distribution. 

## Figures and Tables

**Figure 1 toxins-15-00227-f001:**
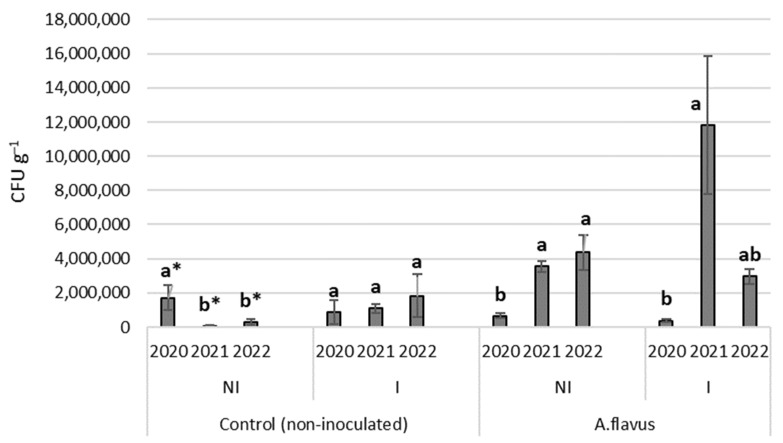
Differences in fungal presence between the examined years and treatments. *p* < 0.05, * *p* < 0.1; different letters indicate significant differences between values; NI—non-irrigated, I—irrigated; vertical lines represent the SD of the mean.

**Figure 2 toxins-15-00227-f002:**
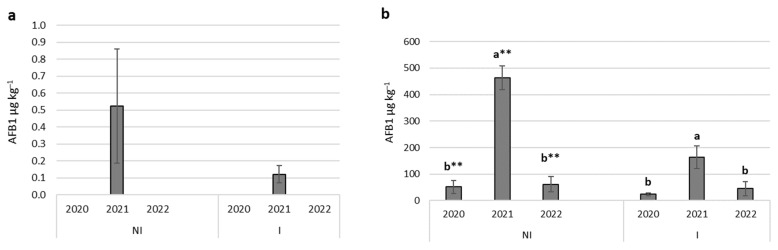
Aflatoxin concentration differences between the examined years in the control (**a**) and inoculated (**b**) treatments. *p* < 0.05, ** *p* < 0.01; different letters indicate significant differences between values; NI—non-irrigated, I—irrigated; vertical lines represent the SD of the mean.

**Figure 3 toxins-15-00227-f003:**
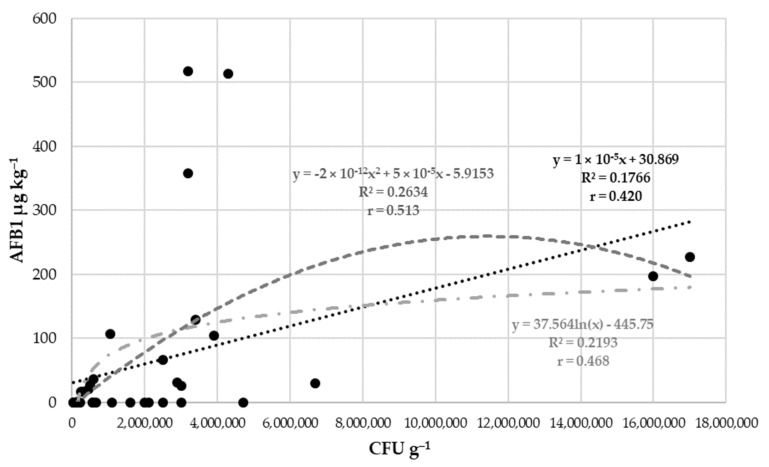
Regression relationships and correlation across years and isolates between mould colony forming units (CFU g^−1^) and aflatoxin B1 concentration in 2020–2022.

**Figure 4 toxins-15-00227-f004:**
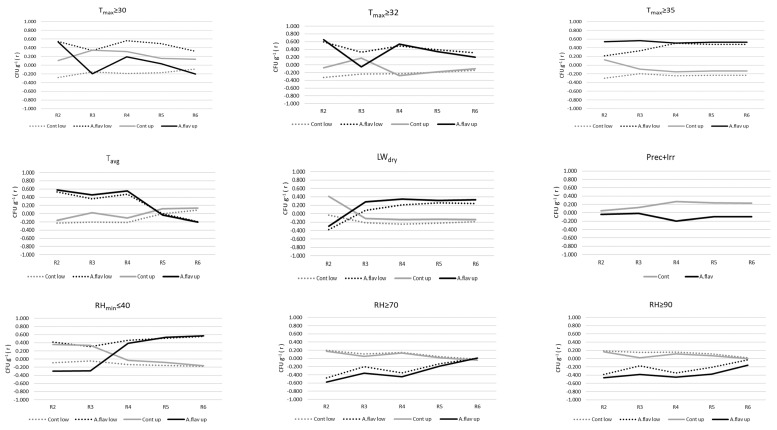
Dynamics of Pearson’s correlation between mould count and environmental factors in the different phenophases, 2020–2022; T_max_ ≥ 30, T_max_ ≥ 32, T_max_ ≥ 35—maximum temperature equal to or higher than 30, 32, and 35 °C, T_avg_—mean daily temperature (°C), LW_dry_—leaf wetness in dry state, Prec+Irr—precipitation and irrigation (mm), RH_min_ ≤ 40—minimum relative humidity equal to or below 40, RH ≥ 70, RH ≥ 90—duration of relative humidity equal to or higher than 70 and 90%; R2—Blister, R3—Milk, R4—Dough, R5—Dent, R6—Physiological maturity; Cont low-Control lower canopy level, *A.flavus* low—Inoculated lower canopy level, Cont up—Control upper canopy level, *A.flavus* up—Inoculated upper canopy level.

**Figure 5 toxins-15-00227-f005:**
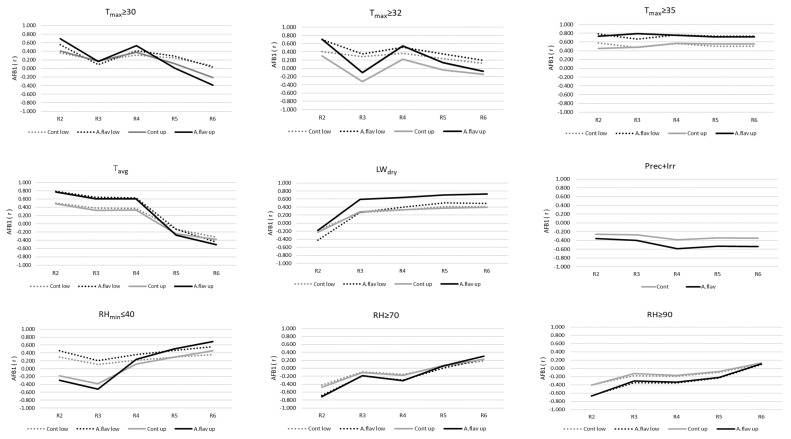
Dynamics of Pearson’s correlation between aflatoxin contamination and environmental factors in the different phenophases, 2020–2022; T_max_ ≥ 30, T_max_ ≥ 32, T_max_ ≥ 35—maximum temperature equal to or higher than 30, 32 and 35 °C, T_avg_—mean daily temperature ( °C), LW_dry_—leaf wetness in dry state, Prec+Irr—precipitation and irrigation (mm), RH_min_ ≤ 40—minimum relative humidity equal to or below 40, RH ≥ 70, RH ≥ 90—duration of relative humidity equal to or higher than 70 and 90%; R2—Blister, R3—Milk, R4—Dough, R5—Dent, R6—Physiological maturity; Cont low—Control lower canopy level, *A.flavus* low-Inoculated lower canopy level, Cont up—Control upper canopy level, *A.flavus* up—Inoculated upper canopy level.

**Table 1 toxins-15-00227-t001:** Analysis of variance of mould colony forming units (CFU g^−1^) under different irrigation in the three-year-long period (2020–2022).

		CFU g^−1^			
Treatment	Irrigation	Mean	SD	Min	Max
Control (non-inoculated)	NI	692,555 b	1,019,322	33,000	3,000,000
	I	1,270,000 a	1,439,237	30,000	4,700,000
*A. flavus* (inoculated)	NI	2,865,555 b	1,904,852	450,000	6,700,000
	I	5,060,000 a	6,288,750	240,000	17,000,000

*p* < 0.05; different letters indicate significant differences between values; NI—non-irrigated, I—irrigated.

**Table 2 toxins-15-00227-t002:** Analysis of variance of aflatoxin B1 contamination under different irrigation treatments in the examined three-year-long period (2020–2022).

		AFB1 µg kg^−1^			
Treatment	Irrigation	Mean	SD	Min	Max
Control (non-inoculated)	NI	0.18 a	0.41	0.00	1.30
	I	0.04 b	0.07	0.00	0.24
*A.flavus* (inoculated)	NI	192.31 a **	201.38	20.28	517.49
	I	77.42 b **	78.91	0.00	227.83

*p* < 0.05, ** *p* < 0.01; different letters indicate significant differences between values; NI—non-irrigated, I—irrigated.

**Table 3 toxins-15-00227-t003:** Seasonal variation of environmental parameters.

	Start Date	End Date	T_avg_	T_max_ ≥ 30	T_max_ ≥ 32	T_max_ ≥ 35	RH_min_ ≤ 40	RH ≥ 70 dur.	RH ≥ 90 dur.	Prec. Sum	LW Dry
**2020**			°C	number of days				day	day	mm	day
**R2**	01. 08.	07. 08.	21.9	3	1	0	2	4.5	2.1	2.2	4.8
**R2–R3**	01. 08.	14. 08.	22.5	10	5	0	4	8.6	4.5	2.2	9.5
**R2–R4**	01. 08.	23. 08.	22.1	13	5	0	4	15.5	9.3	28.4	13.6
**R2–R5**	01. 08.	31. 08.	22.0	16	7	0	8	19.8	11.3	28.6	19.3
**R2–R6**	01. 08.	12. 09.	20.9	17	7	0	12	28.1	16.7	40.8	26.5
**2021**											
**R2**	27. 07	01. 08.	25.7	6	5	1	3	2.6	1.1	2.2	4.6
**R2–R3**	27. 07	09. 08.	23.7	11	6	2	6	7.0	2.9	7.8	10.7
**R2–R4**	27. 07	16. 08.	24.0	17	11	5	13	10.3	4.1	8.8	16.4
**R2–R5**	27. 07	27. 08.	22.3	20	13	5	19	17.2	7.3	24.6	23.7
**R2–R6**	27. 07	12. 09.	20.2	20	13	5	28	26.8	13.6	35.8	32.8
**2022**											
**R2**	27. 07	02. 08.	22.6	5	3	0	3	3.9	2.0	35.0	5.0
**R2–R3**	27. 07	10. 08.	23.0	10	7	1	8	6.2	2.7	35.0	9.3
**R2–R4**	27. 07	17. 08.	23.4	16	10	1	14	8.9	3.2	36.0	13.3
**R2–R5**	27. 07	25. 08.	23.5	21	14	2	17	13.5	5.5	40.2	16.7
**R2–R6**	27. 07	02. 09.	23.2	25	17	2	22	16.8	6.6	60.4	20.9

T_avg_—mean daily temperature, T_max_ ≥ 30, T_max_ ≥ 32, T_max_ ≥ 35—daily maximum temperature equal to or higher than 30, 32, and 35 °C, RH ≤ 40—daily minimum relative humidity equal to or below 40, RH ≥ 70 dur., RH ≥ 90 dur.—duration of relative humidity equal to or higher than 70 and 90%, Prec. sum—Total precipitation, LW_dry_—leaf wetness in dry state, R2-Blister, R3-Milk, R4-Dough, R5-Dent, R6-Physiological maturity stage.

**Table 4 toxins-15-00227-t004:** Results of the multiple regression analysis between colonies (CFU g^−1^) and environmental factors at different canopy levels, 2020–2022.

Treatment	Canopy Level	Phenophase	Model	R	R^2^	Predictors
*A. flavus*	lower	R2	1	0.594	0.353	T_max_ ≥ 32
		R4	1	0.559	0.313	T_max_ ≥ 30
		R5	1	0.511	0.261	RH_min_ ≤ 40
			2	0.677	0.459	RH_min_ ≤ 40, RH ≥ 90
		R6	1	0.556	0.310	RH_min_ ≤ 40
	upper	R2	1	0.647	0.418	T_max_ ≥ 32
		R3	1	0.567	0.321	T_max_ ≥ 35
		R4	1	0.552	0.304	T_avg_
		R5	1	0.538	0.289	RH_min_ ≤ 40
		R6	1	0.574	0.33	RH_and_ ≤40

R2—Blister, R3—Milk, R4—Dough, R5—Dent, R6—Physiological maturity stage.

**Table 5 toxins-15-00227-t005:** Results of the multiple regression analysis between aflatoxin B1 concentration (AFB1 µg kg^−1^) and environmental factors at different canopy levels, 2020–2022.

Treatment	Canopy Level	Phenophase	Model	R	R^2^	Predictors
Control	lower	R2	1	0.574	0.33	T_max_ ≥ 35
		R3	1	0.475	0.226	T_max_ ≥ 35
		R4	1	0.56	0.313	T_max_ ≥ 35
		R5	1	0.502	0.252	T_max_ ≥ 35
		R6	1	0.502	0.252	T_max_ ≥ 35
	upper	R2	1	0.488	0.238	T_avg_
		R3	1	0.483	0.233	T_max_ ≥ 35
		R4	1	0.569	0.324	T_max_ ≥ 35
		R5	1	0.553	0.306	T_max_ ≥ 35
		R6	1	0.553	0.306	T_max_ ≥ 35
*A. flavus*	lower	R2	1	0.787	0.62	T_avg_
			2	0.844	0.712	T_avg_, T_max_ ≥ 35
		R3	1	0.665	0.442	T_max_ ≥ 35
			2	0.823	0.677	T_max_ ≥ 35, Prec+Irr
		R4	1	0.763	0.582	T_max_ ≥ 35
			2	0.834	0.695	T_max_ ≥ 35, Prec+Irr
		R5	1	0.729	0.531	T_max_ ≥ 35
			2	0.829	0.687	T_max_ ≥ 35, Prec+Irr
		R6	1	0.729	0.531	T_max_ ≥ 35
			2	0.833	0.693	T_max_ ≥ 35, Prec+Irr
	upper	R2	1	0.774	0.6	T_avg_
			2	0.834	0.695	T_avg_, RH ≥ 70
		R3 R4	1	0.788	0.622	T_max_ ≥ 35
			1	0.753	0.567	T_max_ ≥ 35
		R5	1	0.72	0.519	T_max_ ≥ 35
		R6	1	0.723	0.522	LW dry

R2—Blister, R3—Milk, R4—Dough, R5—Dent, R6—Physiological maturity stage.

**Table 6 toxins-15-00227-t006:** Technical details of the measurement of climatic parameters.

Parameter	Temperature	Relative Humidity	Leaf Wetness	Precipitation
Sensor type	Onset HOBO RXW-THC-B-868	Onset HOBO RXW-THC-B-868	METER Env. PHYTOS-31	Campbell Scientific ARG-100
Number of sensors	6	6	6	1
Accuracy	±0.2 °C	±2.5% RH	-	±4%
Resolution	0.02 °C	0.01% RH	5 min	0.2 mm/tip
Sampling frequency	10 s	10 s	10 s	-
Logging interval	5 min	5 min	5 min	10 min
Averaging interval	1 day	1 day	1 day (sum)	1 day (sum)
Mounting height	0.3 and 1.5 m	0.3 and 1.5 m	0.3 and 1.5 m	1 m
Data logger applied	Onset Computer Corp. RX3000	Onset Computer Corp. RX3000	Campbell Scientific CR1000	Campbell Scientific CR1000

## Data Availability

Data is contained within the article.

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
