# Peer review of "The Effect of Environmental Factors on Mould Counts and AFB1 Toxin Production by Aspergillus flavus in Maize"

_toxins, 2023, doi:10.3390/toxins15030227_

Round 1

Reviewer 1 Report

Dear author,

I would like to congratulate you for compiling risk factors and considering the impact on A. flavus and aflatoxin contamination rates for three years. However, sometimes these factors  need to be explained and discussed with other researchers to bring to this publication an added value.

It would be interesting to include aflatoxin B1 as a keyword.

 Abstract

Line 16: “ In the case of artificial inoculation, correlations with environmental factors were stronger” Please, it would be better to specify some relevant correlation. Maybe you have to talk something about natural contamination too.

 Introduction

Line 67: It would be interesting to include aflatoxin contamination levels taking into account these conditions in the text and not only indicating the reference.

Line 74 and 75: This affirmation should be explained: “under favourable environmental conditions, A. flavus becomes competitive against other fungi and continues toxin production after harvesting”. We don´t know these favourable conditions and other moulds to be compared”. I do not understand the purpose of this affirmation in the context without information.

The maximum levels laid down in European legislation have not been indicated in the introduction. Perhaps they should be included for later discussion.

Results and discussion

Line 88: Please confirm this affirmation: “mould development was strongly limited by environmental factors”Please, specify to confirm this with results.

Line 89 to 91: Please indicate the units of measurement correctly.

Figure 2a: Maybe you have to explain the EU limits intended for food or feed to choose the destination of the raw materials.

Line 119:  exact p values should be included. Please check it in the text.

Table 5 should be better explained.

Line 221: A. flavus should be written in italics.

Figures 4 and 5 should be better explained in the text. It is not ok to put the figures with a very brief explanation.

Discussion:

Line 252: Please compare Kebede et al results with your results.

I think you should include (highlight) in the discussion the effect of climate change on the presence of aflatoxins in maize under natural conditions.

Author Response

Dear Rewiever,

Thank you for your suggestions and constructive criticism. Please, see our point-by-point responses in the wors text.

Reviewer 2 Report

1. The aim put forward in the introduction is a little abrupt. It needs to reorganize the introduction to make it more logical.

2. Table 1: Why is the SD value so large that it is greater than the mean value? Table 2 has the same problem.

3. Figure 2b: Why does the error bar exceed the limit? Is the error bar drawn by your own hand?

4. Figure 3: The determination coefficient (R2) is less than 0.3, the equation has no significance at all. The determination coefficient in Tab. 4 and Tab. 5 has the same problem.

5. Rather of repeating the findings, the conclusion should sublimate them.

6. Are all data normally distributed?

In general, the article lacks novelty, the reliability of data is questionable, the presented results cannot support the conclusions.

Author Response

Dear Rewiever, 

Thank you for your suggestions and constructive criticism. Please, see our point-by-point responses in the word text.

Reviewer 3 Report

Dear authors, this paper deal with the effect of environmental factors on mould counts and AFB1 toxin production by Aspergillus flavus in maize, interesting topic and very actual. I have some suggestions to improve the paper in the attached file.

Author Response

(The authors gave the same response as above.)

Round 2

Reviewer 1 Report

Dear author,

see below for the addedd comments:

Line 50: Please replace "verticilloides", correct it for "verticillioides" check it along the text.

Line 69: The reference (165/210/EU) is not ok, please, correct it.

Author Response

Dear Reviewer,

We appreciate the time and effort that you dedicated to provide a feedback on our manuscript, authors are grateful for the insightful comments on our present paper. Your thorough evaluation has helped us improve the quality of our research.
Once again, thank you for your invaluable contribution to our work.

Best regards, 

The authors

Reviewer 2 Report

Accept in present form

Author Response

(The authors gave the same response as above.)

Reviewer 3 Report

Dear authors, the paper improved a lot, I endorse the publication

Author Response

(The authors gave the same response as above.)
